# Age, Loneliness, and Social Media Use in Adults during COVID-19: A Latent Profile Analysis

**DOI:** 10.3390/ijerph20115969

**Published:** 2023-05-27

**Authors:** Moira Mckniff, Stephanie M. Simone, Tania Giovannetti

**Affiliations:** Department of Psychology and Neuroscience, Temple University, Philadelphia, PA 19122, USA; moira.mckniff@temple.edu (M.M.); smsimone@temple.edu (S.M.S.)

**Keywords:** age, social media, loneliness, technology barriers, COVID-19

## Abstract

Loneliness has been linked to morbidity and mortality across the lifespan. Social media could reduce loneliness, though research on the relation between social media and loneliness has been inconclusive. This study used person-centered analyses to elucidate the inconsistencies in the literature and examine the possible role technology barriers played in the relation between social media use and loneliness during the COVID-19 pandemic. Participants (*n* = 929; M age = 57.58 ± 17.33) responded to a series of online questions covering demographics, loneliness, technology barriers, and social media use (e.g., Facebook, Twitter, etc.) across a range of devices (e.g., computer, smartphone, etc.). A latent profile analysis was conducted to identify distinct profiles of social media use, loneliness patterns, and age. Results yielded five distinct profiles characterized that showed no systematic associations among age, social media use, and loneliness. Demographic characteristics and technology barriers also differed between profiles and were associated with loneliness. In conclusion, person-centered analyses demonstrated distinct groups of older and younger adults that differed on social media use and loneliness and may offer more fruitful insights over variable-centered approaches (e.g., regression/correlation). Technology barriers may be a viable target for reducing loneliness in adults.

## 1. Introduction

By 2030, 1 in 5 Americans are projected to be over the age of 65 years old [1]. With a rapidly aging population comes the growing public health concern of increased loneliness, as studies have shown that social network size tends to decrease with age [2] and self-perceived loneliness is strongly associated with health outcomes [3]. Social media offers great potential to decrease loneliness, particularly when mobility is limited due to illness or other factors (e.g., physical limitations, COVID-19 lockdown period). However, existing research on the relation between social media use and loneliness has demonstrated mixed results, possibly because most studies have examined the relation between social media use and loneliness using variable-centered statistics (e.g., regression/correlation) separately in younger or older samples. The aim of this study was to use person-centered analyses to identify meaningful groupings of participants based on their age, social media use, and reported loneliness to help provide clarity regarding the equivocal findings across studies using variable-centered approaches.

Social media may offer an avenue for decreasing loneliness, as previous studies have shown that using social media for communicating with friends and family (as opposed to strangers), regularly self-disclosing thoughts and feelings to peers, and sharing frequent status updates to Facebook friends is associated with reduced loneliness [4,5,6]. Connections formed using social media require minimal effort but can occur frequently and between many different people with great efficiency, which can foster a sense of connectedness among users, leading to reduced loneliness [7]. By contrast, Bonsaksen, et al. (2021a) showed that several months into the COVID-19 pandemic, participants aged 18 and older who used social media more frequently (i.e., several times per day) also reported higher levels of loneliness [8,9]. Similarly, in a sample of adults from the US, a majority (73%) of those who described themselves as frequent social media users met criteria for being lonely (UCLA loneliness score at or above 43), whereas only 52% of those who considered themselves to be occasional social media users met criteria for loneliness [10]. Longitudinal studies and theoretical frameworks suggest that problematic reliance on social media increases loneliness [11,12], and in cross sectional studies this relation persists even after controlling for anxiety, depression, and stress [13]. Conflicting results regarding social media use and loneliness may be because the relation between social media use and loneliness may be non-linear, such that *excessive* [14,15] and possibly very infrequent (but not moderate) social media use are associated with increased loneliness.

Age also may explain different results across studies of social media use and loneliness [8,16]. For example, during the first months of the COVID-19 pandemic (April/May 2020), Bonsaksen et al., 2021 showed that in a multinational sample of over 3000 participants, social media use was associated with *lower* levels of loneliness only for middle-aged adult (40–59 years). The relation between social media use and loneliness in younger adults (18–39) or older adults (60+ years) was not statistically significant in separate regression analyses [8]. Several studies have shown no significant association between social media use and loneliness in younger adults [17], but there are conflicting findings for older adults [18,19,20,21]. However, cohort effects in the experience of loneliness could cloud conclusions in samples that include both younger and older people, as Millennials (ages 18 to 34) tend to report more loneliness than people from older generations [22]. Many studies on social media use and loneliness do not include middle-aged adults between the age of 30–65 years, thus, there is a need for clarity regarding the relation between social media use and loneliness in studies that include the full range of adults and consider age as a central factor in the relation between social media and loneliness.

Barriers to accessing or using technology may limit or preclude social media use and are important to understand when considering the potential for social media use to reduce loneliness and may vary across people of different ages. Much of the literature on technology barriers has focused on older adults, as it is hypothesized that they experience more technology barriers relative to younger adults who are more likely to be “digital natives.” A variety of barriers have been identified, including difficulty learning to use technology, errors when using technology, and requiring more assistance with technology [23,24]. In a sample of 400 older adults surveyed during the COVID-19 pandemic, barriers to technology use included general disinterest, lack of access due to both financial limitations and low knowledge, and physical limitations (e.g., poor vision, cognitive impairment) [23]. Low self-efficacy regarding the ability to learn new technologies and anxiety surrounding technology may also prevent older adults from engaging in social media [25]. Trust is another important barrier to older adults’ technology use. News reports about data breeches at popular social media sites, such as Facebook, may dissuade older people from trying or continuing to use social media [25]. Factor analysis has been applied to this research area to identify core constructs in older adults, and four factors have emerged: (1) intrapersonal negative beliefs (e.g., too old, not worth learning), (2) limited knowledge, (3) structural barriers (e.g., cost), and (4) inadequate support (e.g., no one to ask questions to, no one to communicate with about using technology). The most common barriers included cost and dislike towards technology, but more generally, structural and knowledge barriers were the most influential to older adults’ use of technology [24]. Importantly, technology barriers, particularly structural barriers and privacy concerns, can be experienced by individuals of all ages, not just older adults. Therefore, more research is needed to determine if younger people face similar barriers, and if these barriers prevent them from using social media.

As stated, the extant literature, which has relied heavily on variable-centered approaches (e.g., regression), has not demonstrated clear and consistent relations among social media use, loneliness, and age. Additionally, the role of technology barriers in the use of social media to reduce loneliness has not been investigated across the adult lifespan. To address these gaps, this study used person-centered analyses to identify subgroups of adults of various ages who might show different levels of social media use and loneliness during the COVID-19 pandemic when opportunities for socialization outside the home were limited. A person-centered approach could identify subgroups that are similar in age, social media use, and loneliness—a result that is not possible with variable-centered approaches that examine only how variables relate to other variables within the full sample.

Our approach was twofold. First, specific profiles of social media use, loneliness, and age were examined in adults who were 30 years or older during the COVID-19 pandemic. We hypothesized that person-centered analyses would identify specific age groups with distinct profiles on measures of loneliness and social media use. We did not have specific hypotheses about specific features of these subgroups because of the contrasting results reported in the literature to date. However, we anticipated distinct profiles across age groups, with a middle-aged profile of frequent social media users who reported low loneliness [8]. Second, participant subgroups were subsequently examined by their technology barriers, which have been shown to preclude social media use in older adults but have been limitedly studied in younger samples. Comparisons among profiles were conducted to determine whether (and which) technology barriers differed between the profiles. We hypothesized that technology barriers would be negatively associated with social media use. For example, we expected the most technology barriers in groups of older adults who reported loneliness and lower engagement in social media during the COVID-19 pandemic. Demographic features, including, sex, income, education level, race, and ethnicity, were also examined to further characterize the distinct profiles. Results were discussed for their potential to inform strategies to optimize social media features and use for those experiencing loneliness, especially in the context of limited mobility or other external factors that reduce travel outside the home.

## 2. Materials and Methods

This observational study used a cross-sectional design and involved the distribution of an online questionnaire. The study was approved by the Temple University Institutional Review Board. All participants provided informed consent prior to participating in the study.

### 2.1. Participants

Participants were required to be 30 years or older, fluent in English, live in the US or Canada, and have access to a computer, tablet, or smartphone device to complete an online questionnaire. A total of 946 participants responded to the survey. After eliminating participants who failed to pass a manipulation check embedded within the technology barrier questions (“Please select the right-most option [i.e., Strongly agree]”; *n* = 17), the final sample consisted of 929 participants ranging from 30–98 years old. Data were missing for the social media variable (*n* = 5), race (*n* = 4), ethnicity (*n* = 12), and income (*n* = 31). Participant characteristics are summarized in Table 1 and show that the sample included individuals between the ages of 30 to 98 (median = 58.7), the majority of whom were White women. Approximately half of the sample was college educated, with a variety of education levels represented. The sample also contained a range of yearly income, with 34.1% earning less than $35 K per year, 43.1% earning between $35–99 K per year, and 22.8% earning more than $100 K per year.

### 2.2. Procedures

Data were collected online from September 2021 to February 2021 using the Qualtrics web-based survey and recruitment services. Qualtrics relies on an actively managed, double-opt-in market research panel to recruit participants based on designated inclusion/exclusion parameters. We sought an even distribution of adult participants across six 10-year age bands ranging from 30 to 90+ years old. Qualtrics determines compensation for the participants they recruit based on survey length and other factors and the type of reward varies (e.g., cash, gift cards, airline miles, redeemable points, etc.). Our survey included a series of questionnaires that took approximately 15–20 min to complete and assessed demographic information, social media use, loneliness, technology use, and technology barriers.

### 2.3. Measures

#### 2.3.1. Indicator Variables Included in the Latent Profile Analysis

Social Media Use—Social media use was estimated using questions about use of the following digital devices: smartphone, tablet, and personal computer/laptop. For each device that the participant owned and/or used, they were asked to indicate whether they use the device to view or engage in social media at least once per week. Responses collapsed as follows: 0 = no weekly social media use, 1 = uses social media on one device weekly, 2 = uses social media on two devices weekly, 3 = uses social media on three devices weekly. Higher scores were interpreted as reflecting greater social media use and engagement. Frequency of overall device usage was also factored into the overall social media score to estimate the amount of time spent on social media. Participants were asked how frequently they used each device (smartphone, tablet, and computer/laptop). The final social media use score ranged from 0–20, with a score of 0 indicating that the participant did not use social media on any device and a score of 20 indicating that the participant used social media on all of their electronic devices several times per day.

Loneliness—Loneliness was evaluated using the Revised UCLA Loneliness Scale [26] which includes 20 items scored from 1 to 5 (1 = strongly disagree, 2 = disagree, 3 = neutral, 4 = agree, 5 = strongly agree). Some questions were reversed scored. After standardizing all responses, the total score ranges between 20 and 80, with higher scores indicating more loneliness. A median split was used to dichotomize the loneliness variable, with scores under 36 characterized as “not lonely” and scores greater than 36 characterized as “lonely”.

Age—Participants were asked to provide their age in years.

#### 2.3.2. Predictor Variables

Technology Barriers—technology barrier questions consisted of a 25-item survey using a 5-point Likert scale ranging from 1 (‘strongly disagree’) to 5 (‘strongly agree’). Items covered a variety of barriers, such as cost, complexity, limited knowledge, privacy concerns, medical/physical obstacles, and others (see Appendix A). Technology barrier questions were taken from multiple sources, including the technostress questionnaire [27], the Older Adults’ Computer Technology Attitudes Scale (OACTAS) [28], and a focus group on technology barriers in older adult population [29]. A total sum score was computed, ranging from 12 to 125 with higher scores reflecting more technology barriers. Based on the result of a principal component analyses (see Appendix A), items were grouped to measure specific technology barriers (i.e., time/money concerns, limited knowledge, sensory/motor difficulties, and privacy concerns). Because a different number of items was included in each sub-component score, average scores were computed for each component that ranged from 1–5, with higher scores indicating more barriers.

Demographics—In addition to reporting age (indicator carriable described above), participants were asked to indicate their race, ethnicity, sex, education level, and annual income from a series of options. 

### 2.4. Statistical Analyses

Statistical analyses were performed with MPlus Version 5.2 (Los Angeles, CA, USA) [30] and SPSS version 25 (IBM SPSS Statistics for Windows, Armonk, NY, USA). Descriptive analyses were performed for all study variables. Means and standard deviations were used for all variables that were normally distributed. Median and range were used for variables that did not meet normality assumptions. A factor analysis of technology barriers was performed to determine areas where respondents grouped together. Four factors were identified: knowledge, time and money, sensory motor, and privacy. Once items were grouped into a factor, a composite score was created by adding all items with the same factor together. Descriptive statistics of technology barriers are shown in Table 1. Mean averages of each factor within the 5 distinct profiles is shown in Figure 1.

Latent profile analysis (LPA) was used to empirically identify participant groups based on age, social media use, and loneliness scores. LPA was used because of several advantages over other person-centered techniques such as cluster analysis and *k*-means clustering. LPA uses a stepwise procedure and a variety of fit indices to determine whether the addition of classes improves the fit to the data [31]. LPA can accommodate predictor and outcome variables of differing scales and variances, and normally distributed data are not required. LPA assumes that latent profiles are homogenous and that relationships between variables are due to the underlying latent profile; outcomes include model-based, probabilistic classification scores. Mplus default settings were used for the LPA analyses: variances were held equal across classes and covariances among indicators were fixed at zero.

Although there are no “gold standard” guidelines regarding the number of participants and power for a proposed LPA, a variety of statistical indices and conceptual considerations are taken into account to determine which model fits the data best [32,33]. LPA models are fit in a series of steps, starting with a one-profile (independence) model. The number of profiles then is increased one at a time until there is no additional improvement to the fit of the model [32,33]. Statistical fit indices are examined, including the Akaike information criterion (AIC) [34], Bayesian information criterion (BIC) [35], and sample-size adjusted (ABIC) [36]. Lower scores on these indices suggest a better model fit. We also considered the bootstrap likelihood ratio test (BLRT), which examines the fit of the model with *k* profiles to the fit with *k* − 1 profiles [32,33]. Entropy, an index of how well participants fit into distinct profiles that differ from each other, values greater than 0.80 indicate good separation between the identified groups [37]. The size of the smallest profile was also considered; profiles that are too small (e.g., <5–10% of the sample) suggest overfitting of the model to the data, which could limit generalizability. Last, theoretical implications and consistency with conceptual models were also considered. 

Following identification of profiles, between-group differences were examined using Chi-square tests to determine whether profiles significantly differed on demographic variables. A Kruskal–Wallis test was conducted to evaluate differences among the 5 profiles on their total number of technology barriers. Post-hoc tests were then conducted to test pairwise comparisons of total technology barriers between profiles. All analyses were determined to be significant when *p* < 0.05.

## 3. Results

### 3.1. Psychometric Properties of the Study Questionnaires

Cronbach’s *α* and McDonald’s ***ω*** were computed for both the UCLA Loneliness Scale (*α* = 0.94; ***ω*** = 0.94) and the technology barriers total score (*α* = 0.95; ***ω*** = 0.95) and indicated good internal consistency. Internal consistency was not expected for the items that comprise the social media use questionnaire; therefore, Cronbach’s *α* was not computed for this questionnaire.

Items on the technology barriers questionnaire were designed to evaluate specific types of barriers; therefore, principal component analysis (PCA) was conducted to identify sub-scores. Results of the PCA showed four components with good internal reliability representing the following specific barriers: time/money concerns (time and money- *α* = 0.92; ***ω*** = 0.91), limited knowledge (knowledge *α* = 0.93 ***ω*** = 0.93), sensory/motor difficulties (sensory motor; *α* = 0.90; ***ω*** = 0.90), and privacy concerns (privacy; *α* = 0.86; ***ω*** = 0.86). See Appendix A.

Bivariate correlations among the primary measures are reported in Appendix A and supported the validity of the scales. For example, as expected, there were significant, negative associations between the social media use total score and both age and the technology barriers total score. That is, higher social media use scores were associated with younger age and fewer technology barriers. There was no significant bivariate relation between social media use and the UCLA Loneliness score; however, the UCLA Loneliness score was significantly associated with age and technology barriers, such that greater loneliness was associated with older age and more technology barriers.

### 3.2. Latent Profile Analysis

LPA models were conducted using *z*-transformed scores for age, loneliness, and social media use. The one-profile model was a fit first, followed by models with two, three, four, five, and six profiles. As shown in Table 2, multiple model indicators suggested that the five-profile model was the best fit for the data. First, the VLMRLRT and BLRT *p*-values were not statistically significant for the six-profile model. Thus, when considering only the first five models, the AIC, BIC, and ABIC are smallest for the five-profile model. The five-profile model also had good entropy, and the profile with the fewest participants included at least ten percent of the sample (*n* = 99). Finally, the five-profiles were interpretable and conceptually coherent. Therefore, the five-profile model was identified as the final model for interpretation. Mean *z*-scores are shown in Figure 2 to enable comparisons across profiles and variables. Additionally, mean raw scores (and standard deviations) for age, loneliness, and social media use across each of the five profiles, along with their characterizations, are included in Table 3. 

### 3.3. Demographic Differences among the Profiles

Omnibus chi-square tests were performed to examine demographic differences among the profiles. To control for type I error, only omnibus tests with *p*-values < 0.01 were considered meaningful and further investigated with post-hoc chi-square tests. Results of the omnibus tests are shown in Table 4. As seen in Table 4, post-hoc tests showed that the profiles statistically differed on several demographic variables with a medium effect size for all comparisons, except ethnicity (small effect). Results can be summarized as follows: (1) profile 5 included significantly fewer men; (2) profile 2 included more participants with greater education; (3) profiles 1 and 2 included more participants with higher annual incomes, whereas profile 4 included more participants with lower annual incomes; (4) profile 5 included more participants that identified as races other than White.

### 3.4. Technology Barriers

A Kruskal–Wallis test was conducted to evaluate differences among the five profiles on their total number of technology barriers (see Table 4). Post-hoc pairwise comparisons showed that profile 2 had the lowest total technology barriers score compared to all other profiles. Profile 5 also had a significantly lower total technology barriers score than profiles 1, 3, and 4. There were no other significant differences between the profiles. The results of post-hoc between group statistical comparisons are included in Appendix A).

Mean scores for each of the technology barriers component scores were examined qualitatively. As shown in Figure 1, privacy concerns and limited knowledge were consistently reported as greater barriers than time/money and sensory/motor limitations. Knowledge barriers were higher in the older profiles (e.g., profiles 3 and 4).

## 4. Discussion

Person-centered analyses showed five profiles of adults that differed in age, social media use, and loneliness. Profiles that reported high levels of loneliness (profiles 1 and 4) also reported average social media use, and profiles that did not report feeling lonely (profiles 2, 3, and 5) showed all levels of social media use (frequent, average, infrequent). When considering age, among the younger adults in their late 30s (profiles 1 and 2), more frequent social media use was associated with lower levels of loneliness, but this pattern was not observed for older participants. For example, we identified a subgroup of lonely older adults in their late 60’s (post-retirement age) who used social media more than a subgroup of the oldest adults in their 80’s that reported low levels of loneliness. Thus, our findings, along with the equivocal findings reported in the literature to date suggest person-centered analyses are highly suitable for identifying meaningful profiles of participants with unique patterns of social media use, age, and loneliness. Variable-centered analyses and any approach that assumes linear relations among social media use, age, and loneliness will likely continue to yield conflicting results and obscure meaningful patterns among subgroups of people, particularly among adults from different age groups who may be in very different life-stages and social roles with very different demands.

Demographic features, including, sex, income, education level, race, and ethnicity and technology barriers were examined to further characterize the distinct profiles. The profiles with the highest loneliness did not possess similar demographic characteristics, but they were comparable in terms of technology barriers. Profiles 1 and 4 (participants with more loneliness), as well as Profile 3 (oldest participants), reported more technology barriers than people in the other profiles. Among both the middle-aged and older adult age groups (i.e., people in their late 30s and late 60s), participants who reported more technology barriers also reported greater loneliness. When considering specific technology barriers, our results showed that limited knowledge and privacy concerns were most relevant for participants in every profile. These specific barriers might be targeted to reduce loneliness. Although speculative, it is possible that removing technology barriers may offer people who feel lonely ways to use technology to (1) engage with people through avenues that do not involve social media (i.e., learning about community events, etc.) or (2) access non-social activities that are stimulating and indirectly reduce loneliness.

As stated, our results show no clear and consistent relation between frequent social media use and decreased loneliness among all adults. However, our measure of social media use was general and did not specify specific platforms or social media use behaviors, which may have specific relations with loneliness. For example, Facebook usage was associated with an increase in relationship tie-strength more than face-to-face communication in a sample of 3649 Facebook users. These effects were seen in both direct forms of Facebook use (posting on walls, messaging with friends and family members, commenting on posts) and indirect forms (reading a friend’s status updates) [38]. However, Phu and Gow (2019) found that more persistent usage of Facebook was associated with greater loneliness, and a larger number of friends on Facebook predicted higher levels of loneliness in younger adults in their 20’s [39]. With respect to Instagram, using the app to browse and interact with content posted by friends and peers was related to lower levels of loneliness, whereas posting content without interacting with posts made by others (i.e., broadcasting) was related to higher levels of loneliness [40]. These findings suggest that social media use is multi-faceted and nuanced, with particular platforms and specific behaviors differentially contributing to social connectedness and loneliness. Though our social media use measure did not assess the use of specific platforms or specific social media behaviors, future work should do so using person-centered analyses.

We acknowledge several limitations and highlight strengths of our study. First, the sample was racially and ethnically homogeneous and comprised mostly of Non-Hispanic White individuals, which limits the generalizability of our results. Additionally, our measure of social media use was developed for this study and has not been extensively studied. However, results from bivariate correlations support the validity of the measure as it was predictably (and significantly) associated with age and technology barriers (see Appendix A). Additionally, asking questions about barriers to technology use through an online questionnaire may have resulted in a biased sample. Respondents in this study had to have some familiarity with technology in order to complete the questionnaire. It is possible this study did not capture extremely lonely adults who do not use social media because they have barriers that prevent them from even accessing or using any technology at all. These limitations notwithstanding, our study also had numerous strengths that served to address gaps in the extant literature. For example, our use of person-centered analyses is novel and conceptually advantageous to map the complexities among social media use, loneliness, and age. Additionally, we recruited a large sample that provided more than sufficient power to detect effects in all of our study analyses. Finally, to our knowledge, this study is one of few that examine technology barriers in a sample that includes people in late young adulthood and middle age (i.e., 30–65 years old). As such, our novel findings indicate that technology barriers can impact adults of all ages and can influence subjective experiences of loneliness regardless of social media use.

## 5. Conclusions

In conclusion, person-centered analysis proved to be useful in identifying subgroups of adults with different patterns of social media usage and loneliness during the COVID-19 pandemic. Across the five profiles, social media use was not clearly and directly linked to loneliness. Though, among profiles of younger middle-aged adults in their late 30s and early 40s (profiles 1 and 2), more frequent social media use was associated with lower levels of loneliness, but this pattern was not observed for older participants. Additionally, higher technology barriers were associated with higher levels of loneliness across the five profiles. Results support the importance of addressing technology barriers, particularly privacy concerns and limited technology knowledge across all ages, which may work to alleviate technology barriers as well as loneliness. Future studies should incorporate person-centered analyses to examine specific aspects of social media use (frequency, type of social media platform) and their effects on loneliness across the lifespan.

## Figures and Tables

**Figure 1 ijerph-20-05969-f001:**
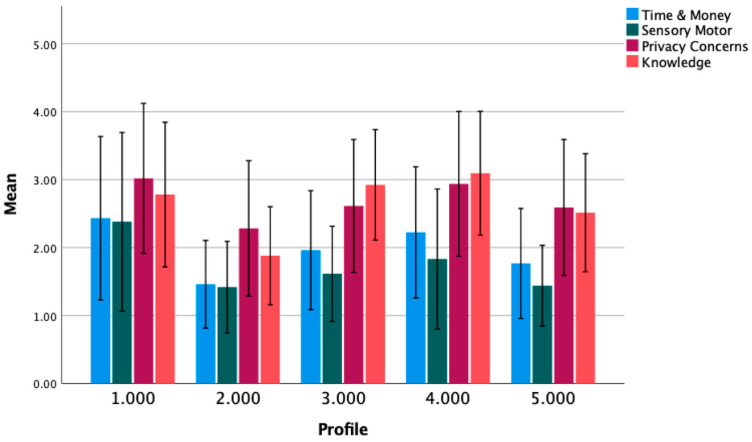
Average technology barriers per profile. Error bars reflect ±1 standard deviation.

**Figure 2 ijerph-20-05969-f002:**
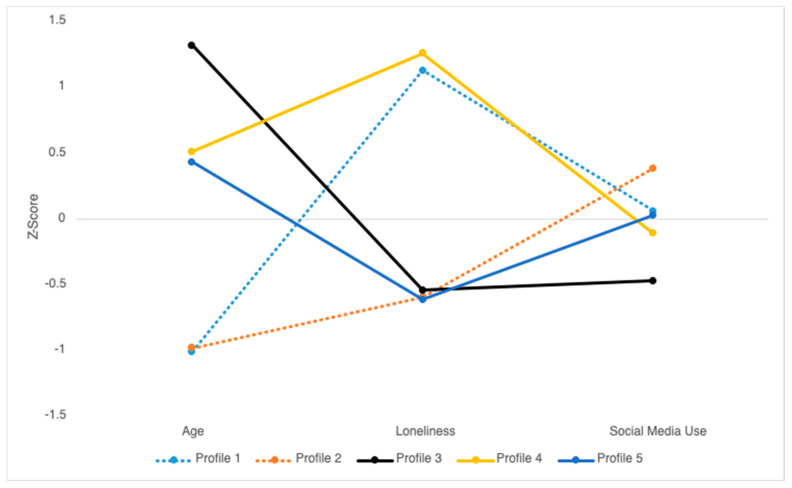
*Z*-transformed distributions of age, loneliness, and social media use across the 5 profiles.

**Table 1 ijerph-20-05969-t001:** Demographic characteristics of sample and average scores on indicator and predictor variables (*n* = 929).

Demographic Characteristic	Mdn (Range) or % (*n*)
Age	58 (30–97)
Sex (% women)	61.2% (*n* = 569)
Education	---
% High School Degree	22.8% (*n* = 212)
% College Degree	51.2% (*n* = 476)
% Graduate Degree	25.9% (*n* = 241)
Annual Income	---
<$35 K per year	34.1% (*n* = 306)
$35–99 K per year	43.1% (*n* = 387)
>$100 K per year	22.8% (*n* = 205)
Race (% White)	90.2% (*n* = 834)
Ethnicity (% Hispanic/Latino)	2.9% (*n* = 27)
**Indicator Variables**	**Mean (SD)**
Social Media Use	7.81 (6.04)
UCLA Loneliness	38.21 (12.78)
**Predictor Variables**	**Mean (SD)**
Technology Barriers	59.90 (20.15)
Knowledge	25.86 (9.68)
Time and Money	9.71 (4.87)
Sensory Motor	5.16 (2.88)
Privacy Concerns	10.68 (4.23)

**Table 2 ijerph-20-05969-t002:** Fit Indices for LPA Models with 1–6 Profiles.

# of Profiles	Free Parameters	AIC	BIC	ABIC	VLMRLRT *p*-Value	BLRT *p*-Value	Entropy	*n* of Smallest Group	% of Sample
1	6	7903.97	7932.98	7913.92					
2	10	7630.48	7678.82	7647.06	0.0000	0.0000	0.80	448	0.48
3	14	7578.60	7646.28	7601.81	0.0000	0.0000	0.78	194	0.21
4	18	7517.02	7604.03	7546.87	0.0001	0.0001	0.77	93	0.10
5	22	7480.77	7587.12	7517.25	0.0318	0.0356	0.75	99	0.11
6	26	6149.66	6275.35	6192.78	0.0577	0.0627	0.93	42	0.05

AIC = Akaike information criterion; BIC = Bayesian information criterion; ABIC = sample-size adjusted BIC; BLRT = bootstrap likelihood ratio test.

**Table 3 ijerph-20-05969-t003:** Mean raw scores (and standard deviations) of primary variables and characterizations for profiles 1–5.

	Profile 1 (*n* = 210)	Profile 2 (*n* = 180)	Profile 3 (*n* = 164)	Profile 4 (*n* = 99)	Profile 5 (*n* = 276)
Age	39.69 (8.81)	40.30 (5.84)	82.24 (5.49)	67.01 (7.78)	64.44 (5.91)
Loneliness	52.71 (7.49)	30.69 (6.17)	31.27 (7.44)	54.33 (7.28)	30.43 (7.17)
Social Media Use	8.12 (5.90)	10.42 (6.05)	4.88 (5.55)	6.92 (5.71)	7.96 (5.74)
Characterization	Younger adults	Younger adults	Younger adults	Older adults	Middle-aged adults
	Lonely	Not lonely	Not lonely	Lonely	Not lonely
	Average social media use	Frequent social media use	Infrequent social media use	Average social media use	Average social media use

**Table 4 ijerph-20-05969-t004:** Demographic information and between-group differences for profiles 1–5.

	Profile 1	Profile 2	Profile 3	Profile 4	Profile 5	Omnibus Chi Square or Kruskal–Wallis	*df*	Effect Size/Cramer ’s V or Eta^2^
Sex (% Woman)	56 ^a^	56 ^a^	57 ^a^	66 ^a,b^	70 ^b^	16.125 *	4	0.132 (medium)
Education	-	-	-	-	-	30.684 *	8	0.129 (medium)
% High School Degree	23 ^a^	12 ^b^	30 ^a^	28 ^a^	24 ^a^	-	-	-
% College Degree	47 ^a^	53 ^a^	47 ^a^	56 ^a^	54 ^a^			
% Graduate Degree	30.5 ^a,b^	35 ^b^	23 ^a,b^	16 ^a^	22 ^a^			
Income	-	-	-	-	-	45.918 *	8	0.160 (medium)
<$35 K per year [%]	37.5 ^a,b^	23 ^c^	39 ^a,b^	52 ^b^	30 ^a,c^	-	-	-
$35–99 K per year [%]	33 ^a^	46 ^a,b^	44 ^a,b^	36 ^a,b^	51 ^b^	-	-	-
>$100 K per year [%]	30 ^a^	30.5 ^a^	17 ^a,b^	13 ^b^	19 ^a,b^	-	-	-
Race (% White)	80 ^a^	90 ^a,b^	97 ^b^	94 ^b^	93 ^b^	34.829 *	4	0.194 (medium)
Ethnicity (% Hispanic/Latino)	4	6	2.5	2	1	9.175	4	0.100 (small)
Technology Barriers M (SD)	67.82 ^a^ (23.81)	46.92 ^b^ (14.98)	63.38 ^a^ (17.14)	68.31 ^a^ (18.84)	57.27 ^c^ (17.18)	129.96 *	4	0.136 (medium)

Values in the columns with different superscript letters differ significantly (*p* < 0.05). * *p* < 0.01.

## Data Availability

The data presented in this study are openly available in Open Science Framework Storage at https://doi.org/10.17605/OSF.IO/R3Z72.

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
