# Peer review of "Age, Loneliness, and Social Media Use in Adults during COVID-19: A Latent Profile Analysis"

_ijerph, 2023, doi:10.3390/ijerph20115969_

Round 1

Reviewer 1 Report

The manuscript reports the findings of a latent profile analysis (LPA) of age, loneliness and social media use in a sample of adults. The findings are interesting and informative, in the way that various distinct patterns of the levels of these variables (i.e., profiles) were identified. The recruitment and statistical analysis were clearly presented and well justified. These profiles were differentiated by the levels of technology barriers, offering insights into the role of perceived barriers in the distinct profiles of loneliness, social media and age. I have a few minor comments for the authors’ consideration, as follows:

Introduction:

-          The authors suggested mixed results concerning the relationship between social media use and loneliness. In addition to evidence supporting the negative association that is already cited in Paragraph 2 (lines 48 – 59), the authors can cite evidence for the positive association between loneliness and network use (e.g. network use facilitates maintenance of loneliness or lonelier individuals tend to use social media more often), to have a balanced discussion on the issue of "mixed evidence". This would better inform the use of latent profile analysis to identify distinct patterns of network use and loneliness.

-          The Introduction should be beefed up with more discussion of the influence of age on the relationship between social media use and loneliness. This would justify including age as an indicator variable in the LPA. Rather, it is more common to consider age as a predictor of the profiles.

-          The meaning of the statement “Several studies have reported different results even within the same sample” (line 60, p. 2) is unclear. Does it mean the relationship between network use and loneliness is weak (maybe due to the presence of distinct profiles sp the relationships were masked), or do different results arise from various statistical or methodological decisions? Please clarify or revise the statement.

-          The rationale for the use of a person-centred approach to understand the relationship between age, loneliness, and network use is well explained in the last paragraph of the Introduction. I also understand the justification for the lack of specific hypotheses (lines 107-111, p. 3). I would recommend making some speculations about the number of profiles and/ or the description of the profiles by the patterns of levels of age, loneliness and network use, based on the literature reviewed earlier.

Methods:

-          It is clearer if the measures can be divided into two subsections: indicator variables of the LPA and predictors (e.g. demographics and technology barriers).

-          Some details of the LPA, especially on the specification of the variance/ covariance structure between profiles (e.g. homogenous variance-covariance matrix across profiles? Covariance between indicator variables set to zero or freely estimated?), should be mentioned.

Results:

-          The characterization of the profiles is not very straightforward. In particular, it is hard to tell if the levels of network use were “average”, “frequent”, or “infrequent”, and if the levels of loneliness were “lonely” or “not lonely”. I recommend conducting an ANOVA analysis and subsequent post-hoc analysis on the levels of these indicator variables by profiles. The results will give a clearer picture of the characteristics of these profiles and labels of the levels.

-          The internal consistency of the factors of the technology barrier questionnaire, which is a novel measure, could be reported for the reader’s reference.

-          For section 3.5, a table showing the results of the Kruskal-Wallis test and the post-hoc analyses should be added. These results would be informative so readers will know more about the comparisons of the levels of technology barriers and the underlying factors across profiles.

Discussion:

-          The current results didn’t say much about the linear relationship between social media use and loneliness (line 309 – 310, p. 9), at least the sample as a whole. But it may give a clue to the relationship within the profile, as it is pre-specified in the model (again, more details on the specification of the LPA should be given, see my second comment under Methods). A revision of that statement is suggested.

-          The use of causal language (e.g. Our results suggest that social media use alone is not sufficient for reducing loneliness among adults, line335 – 336, p. 9) should be avoided, as this sample is cross-sectional.

Author Response

Dear IJERPH Editors:  

Thank you for the prompt and comprehensive review of our paper, Age, Loneliness, and Social Media Use in Adults During COVID-19: A Latent Profile Analysis.” We were pleased and greatly appreciate the many complimentary remarks from the reviewers. Additionally, we have carefully considered the Reviewers comments and have revised the original manuscript accordingly. Below, we list each of the Reviewers’ critiques and state how we addressed each point in our revision (in bold). Changes in the manuscript are indicated by red font 

REVIEWER 1  

Comments and Suggestions for Authors 

The manuscript reports the findings of a latent profile analysis (LPA) of age, loneliness and social media use in a sample of adults. The findings are interesting and informative, in the way that various distinct patterns of the levels of these variables (i.e., profiles) were identified. The recruitment and statistical analysis were clearly presented and well justified. These profiles were differentiated by the levels of technology barriers, offering insights into the role of perceived barriers in the distinct profiles of loneliness, social media and age. I have a few minor comments for the authors’ consideration, as follows: 

Introduction: 

  • The authors suggested mixed results concerning the relationship between social media use and loneliness. In addition to evidence supporting the negative association that is already cited in Paragraph 2 (lines 48 – 59), the authors can cite evidence for the positive association between loneliness and network use (e.g. network use facilitates maintenance of loneliness or lonelier individuals tend to use social media more often), to have a balanced discussion on the issue of "mixed evidence". This would better inform the use of latent profile analysis to identify distinct patterns of network use and loneliness.  

RESPONSE: We have added three additional citations that demonstrate the positive association between excessive social media use and loneliness: Marttila, et al. 2021; Moretta & Buodo (2020); Youssef, et al. 2020 in the revised Introduction.  

  • The Introduction should be beefed up with more discussion of the influence of age on the relationship between social media use and loneliness. This would justify including age as an indicator variable in the LPA. Rather, it is more common to consider age as a predictor of the profiles.  

RESPONSE: We have added a new paragraph to add depth to the background of the role of age in the relation between social media use and loneliness. Several new references have been added as well as our rationale for including age in the LPA.  

  • The meaning of the statement “Several studies have reported different results even within the same sample” (line 60, p. 2) is unclear. Does it mean the relationship between network use and loneliness is weak (maybe due to the presence of distinct profiles so the relationships were masked), or do different results arise from various statistical or methodological decisions? Please clarify or revise the statement.  

RESPONSE: We have edited the Introduction for clarity and added details to clarify the confusing statement identified by the reviewer.   

  • The rationale for the use of a person-centred approach to understand the relationship between age, loneliness, and network use is well explained in the last paragraph of the Introduction. I also understand the justification for the lack of specific hypotheses (lines 107-111, p. 3). I would recommend making some speculations about the number of profiles and/ or the description of the profiles by the patterns of levels of age, loneliness and network use, based on the literature reviewed earlier.  

RESPONSE: We have edited the Introduction to include the prediction that we would identify a group of middle-aged adults who reported low loneliness and high social media use consistent with the results from Bonsaksen et al., 2021.  

Methods: 

  • It is clearer if the measures can be divided into two subsections: indicator variables of the LPA and predictors (e.g. demographics and technology barriers). 

RESPONSE: We have edited the Methods section to include Headers for Indicator Variables and Predictor Variables.  

  • Some details of the LPA, especially on the specification of the variance/ covariance structure between profiles (e.g. homogenous variance-covariance matrix across profiles? Covariance between indicator variables set to zero or freely estimated?), should be mentioned. 

RESPONSE: We have added this information to section 2.4 Statistical Analyses in the revised paper (i.e., equal variances and covariances fixed to zero). 

Results: 

  • The characterization of the profiles is not very straightforward. In particular, it is hard to tell if the levels of network use were “average”, “frequent”, or “infrequent”, and if the levels of loneliness were “lonely” or “not lonely”. I recommend conducting an ANOVA analysis and subsequent post-hoc analysis on the levels of these indicator variables by profiles. The results will give a clearer picture of the characteristics of these profiles and labels of the levels.  

RESPONSE: We agree that the characterization of the profiles was not clear. We conducted the ANOVAs as suggested, and not surprisingly, they were all statistically significant (p <.001 for all three ANOVAs). See below:  

Tests of Between-Subjects Effects for Age 

Dependent Variable:   Age in yrs   

Source 

Type III Sum of Squares 

df 

Mean Square 

F 

Sig. 

Corrected Model 

807.345a 

4 

201.836 

1545.702 

<.001 

Intercept 

3.664 

1 

3.664 

28.057 

<.001 

Class 

807.345 

4 

201.836 

1545.702 

<.001 

Error 

120.655 

924 

.131 

Total 

928.000 

929 

Corrected Total 

928.000 

928 

a. R Squared = .870 (Adjusted R Squared = .869) 

Tests of Between-Subjects Effects for Social Media Composite 

Dependent Variable:   SocialMediaComposite 

Source 

Type III Sum of Squares 

df 

Mean Square 

F 

Sig. 

Corrected Model 

74.422a 

4 

18.605 

20.149 

<.001 

Intercept 

.485 

1 

.485 

.526 

.469 

Class 

74.422 

4 

18.605 

20.149 

<.001 

Error 

848.578 

919 

.923 

Total 

923.000 

924 

Corrected Total 

923.000 

923 

a. R Squared = .081 (Adjusted R Squared = .077) 

Tests of Between-Subjects Effects for Loneliness 

Dependent Variable:   Loneliness   

Source 

Type III Sum of Squares 

df 

Mean Square 

F 

Sig. 

Corrected Model 

640.942a 

4 

160.235 

515.775 

<.001 

Intercept 

14.235 

1 

14.235 

45.821 

<.001 

Class 

640.942 

4 

160.235 

515.775 

<.001 

Error 

287.058 

924 

.311 

Total 

928.000 

929 

Corrected Total 

928.000 

928 

a. R Squared = .691 (Adjusted R Squared = .689) 

Post-hoc comparisons supported our characterization of the profiles with respect to social media use and loneliness.  For example, for social media use there was no significant difference among the profiles labeled as “average”, though the three “average” profiles obtained significantly lower ratings than the profile labeled as “high average” and significantly higher ratings than the profile labeled as “low average.”  See below:   

Multiple Comparisons for Social Media Use 

Dependent Variable:   SocialMediaComposite   

Bonferroni   

(I) Class 

(J) Class 

Mean Difference (I-J) 

Std. Error 

Sig. 

95% Confidence Interval 

Lower Bound 

Upper Bound 

1.000 

2.000 

-.3821091* 

.09820100 

.001 

-.6584300 

-.1057882 

3.000 

.5357771* 

.10013651 

<.001 

.2540100 

.8175441 

4.000 

.1987248 

.11714944 

.902 

-.1309137 

.5283634 

5.000 

.0269444 

.08806090 

1.000 

-.2208440 

.2747327 

2.000 

1.000 

.3821091* 

.09820100 

.001 

.1057882 

.6584300 

3.000 

.9178862* 

.10429163 

<.001 

.6244273 

1.2113450 

4.000 

.5808339* 

.12072040 

<.001 

.2411473 

.9205205 

5.000 

.4090535* 

.09275852 

<.001 

.1480468 

.6700601 

3.000 

1.000 

-.5357771* 

.10013651 

<.001 

-.8175441 

-.2540100 

2.000 

-.9178862* 

.10429163 

<.001 

-1.2113450 

-.6244273 

4.000 

-.3370523 

.12230004 

.060 

-.6811837 

.0070792 

5.000 

-.5088327* 

.09480522 

<.001 

-.7755984 

-.2420670 

4.000 

1.000 

-.1987248 

.11714944 

.902 

-.5283634 

.1309137 

2.000 

-.5808339* 

.12072040 

<.001 

-.9205205 

-.2411473 

3.000 

.3370523 

.12230004 

.060 

-.0070792 

.6811837 

5.000 

-.1717805 

.11262637 

1.000 

-.4886919 

.1451309 

5.000 

1.000 

-.0269444 

.08806090 

1.000 

-.2747327 

.2208440 

2.000 

-.4090535* 

.09275852 

<.001 

-.6700601 

-.1480468 

3.000 

.5088327* 

.09480522 

<.001 

.2420670 

.7755984 

4.000 

.1717805 

.11262637 

1.000 

-.1451309 

.4886919 

Based on observed means. 

 The error term is Mean Square(Error) = .923. 

*. The mean difference is significant at the 0.05 level. 

Post-hoc for the UCLA loneliness scale also showed a significant difference between the two profiles labeled as “lonely” versus all others. See below:  

Multiple Comparisons for UCLA Loneliness Score 

Dependent Variable:   Loneliness   

Bonferroni   

(I) Class 

(J) Class 

Mean Difference (I-J) 

Std. Error 

Sig. 

95% Confidence Interval 

Lower Bound 

Upper Bound 

1.000 

2.000 

-1.7229743* 

.05661553 

<.001 

-1.8822786 

-1.5636699 

3.000 

-1.6780630* 

.05808358 

<.001 

-1.8414981 

-1.5146278 

4.000 

.1270848 

.06795183 

.618 

-.0641176 

.3182872 

5.000 

-1.7432963* 

.05103911 

<.001 

-1.8869098 

-1.5996828 

2.000 

1.000 

1.7229743* 

.05661553 

<.001 

1.5636699 

1.8822786 

3.000 

.0449113 

.06016864 

1.000 

-.1243908 

.2142134 

4.000 

1.8500591* 

.06974248 

<.001 

1.6538182 

2.0463000 

5.000 

-.0203220 

.05339994 

1.000 

-.1705784 

.1299344 

3.000 

1.000 

1.6780630* 

.05808358 

<.001 

1.5146278 

1.8414981 

2.000 

-.0449113 

.06016864 

1.000 

-.2142134 

.1243908 

4.000 

1.8051478* 

.07093940 

<.001 

1.6055390 

2.0047565 

5.000 

-.0652333 

.05495397 

1.000 

-.2198624 

.0893958 

4.000 

1.000 

-.1270848 

.06795183 

.618 

-.3182872 

.0641176 

2.000 

-1.8500591* 

.06974248 

<.001 

-2.0463000 

-1.6538182 

3.000 

-1.8051478* 

.07093940 

<.001 

-2.0047565 

-1.6055390 

5.000 

-1.8703811* 

.06529691 

<.001 

-2.0541131 

-1.6866491 

5.000 

1.000 

1.7432963* 

.05103911 

<.001 

1.5996828 

1.8869098 

2.000 

.0203220 

.05339994 

1.000 

-.1299344 

.1705784 

3.000 

.0652333 

.05495397 

1.000 

-.0893958 

.2198624 

4.000 

1.8703811* 

.06529691 

<.001 

1.6866491 

2.0541131 

Based on observed means. 

 The error term is Mean Square(Error) = .311. 

*. The mean difference is significant at the 0.05 level. 

Pos-hoc analyses for age showed more significant differences between the groups we distinguished as “younger” versus “older” and the “oldest” profile was significantly older than all other groups. Analyses also showed significant differences between the two groups that we had originally characterized as “older.”  See below. These analyses made us reconsider our original age-labels. Consequently, we edited Table 3 and the profile labels throughout he paper, to improve the clarity of the profile age, classifications.  

Multiple Comparisons for Age 

Dependent Variable:   age in yrs   

Bonferroni   

(I) Class 

(J) Class 

Mean Difference (I-J) 

Std. Error 

Sig. 

95% Confidence Interval 

Lower Bound 

Upper Bound 

1.000 

2.000 

-.6198 

.63609 

1.000 

-2.4097 

1.1700 

3.000 

-42.5521* 

.65259 

<.001 

-44.3883 

-40.7158 

4.000 

-27.3244* 

.76346 

<.001 

-29.4726 

-25.1762 

5.000 

-24.7563* 

.57344 

<.001 

-26.3699 

-23.1428 

2.000 

1.000 

.6198 

.63609 

1.000 

-1.1700 

2.4097 

3.000 

-41.9322* 

.67602 

<.001 

-43.8344 

-40.0301 

4.000 

-26.7045* 

.78358 

<.001 

-28.9094 

-24.4997 

5.000 

-24.1365* 

.59997 

<.001 

-25.8247 

-22.4483 

3.000 

1.000 

42.5521* 

.65259 

<.001 

40.7158 

44.3883 

2.000 

41.9322* 

.67602 

<.001 

40.0301 

43.8344 

4.000 

15.2277* 

.79703 

<.001 

12.9850 

17.4704 

5.000 

17.7958* 

.61743 

<.001 

16.0585 

19.5331 

4.000 

1.000 

27.3244* 

.76346 

<.001 

25.1762 

29.4726 

2.000 

26.7045* 

.78358 

<.001 

24.4997 

28.9094 

3.000 

-15.2277* 

.79703 

<.001 

-17.4704 

-12.9850 

5.000 

2.5681* 

.73363 

.005 

.5038 

4.6324 

5.000 

1.000 

24.7563* 

.57344 

<.001 

23.1428 

26.3699 

2.000 

24.1365* 

.59997 

<.001 

22.4483 

25.8247 

3.000 

-17.7958* 

.61743 

<.001 

-19.5331 

-16.0585 

4.000 

-2.5681* 

.73363 

.005 

-4.6324 

-.5038 

Based on observed means. 

 The error term is Mean Square(Error) = 39.217. 

*. The mean difference is significant at the 0.05 level. 

Although ANOVAs and post-hoc tests are interesting to consider when characterizing the profiles, we are reluctant to include the ANOVA results in the modified paper to keep the focus on the person-centered analyses. The Profiles are the result of more than simple between group differences and also consider the relations among indicators within each profile. Thus, the use of ANOVA runs counter to the person-centered approach that is a key feature of the paper.  

  • The internal consistency of the factors of the technology barrier questionnaire, which is a novel measure, could be reported for the reader’s reference.  

RESPONSE: Alphas were computed for each of the factors from the Technology Barriers questionnaire and are reported in the section, “Psychometric Properties of the Study Questionnaires.” 

  • For section 3.5, a table showing the results of the Kruskal-Wallis test and the post-hoc analyses should be added. These results would be informative so readers will know more about the comparisons of the levels of technology barriers and the underlying factors across profiles. 

RESPONSE: The results of the Kruskal-Wallis test for the total Technology Barriers questionnaire is included in the last line of Table 4. Results of the post-hoc comparisons were included in the Supplement.  

Discussion:  

  • The current results didn’t say much about the linear relationship between social media use and loneliness (line 309 – 310, p. 9), at least the sample as a whole. But it may give a clue to the relationship within the profile, as it is pre-specified in the model (again, more details on the specification of the LPA should be given, see my second comment under Methods). A revision of that statement is suggested. 

RESPONSE: No relations were pre-specified within the model (see point above). We have edited the Discussion section to be clearer about what the LPA results suggest regarding the relation between social media use and loneliness. 

  • The use of causal language (e.g. Our results suggest that social media use alone is not sufficient for reducing loneliness among adults, line335 – 336, p. 9) should be avoided, as this sample is cross-sectional. 

RESPONSE: We have corrected this error in the revised Discussion section. We also reviewed the entire manuscript with this point in mind and made some additional edits throughout. 

Reviewer 2 Report

Comments and suggestions for Authors

I read with interest the manuscript entitledAge, Loneliness, and Social Media Use in Adults During COVID-19: A Latent Profile Analysis’.

I appreciate the authors' idea to approach the relationship between loneliness, age and the use of social media using person-centered analysis.

The manuscript is presented in an organized, systematized manner.

The introduction is clear. However, the authors should structure it:

-          the paragraph referring to the aim of the study (lines 39-42) should be written after the literature review and related with paragraphs about the study’s approach (line 102) and the hypothesis (line 107, 114);

-          the information about participants (lines 42-47) should be also written after the presentation of previous studies and possible related to the paragraph referring to the variables included in the study (lines 111).

Regarding the method, the statistical analysis is elaborated, and the identification of the models five profiles of adults that differed in age, social media use, and loneliness is interesting.

In the results section,

-          table 1 – the name of the second column should perhaps be changed, not everything written below represents means and SD (there are also percentages, numbers);

-          table 4 –the specification regarding the letters a, b, c is not clear: according to the legend of the table, it would seem that all values are significant at p< 0.05 and it is not clear why some letters are written in normal font (comparing with the ones written in superscript format)

Discussions are clearly presented.

I congratulate the authors that, through the person-centered approach and identification of the 5 profiles, they emphasized that the relationship between age, loneliness and the use of social media must be viewed from a more nuanced perspective.

Date: 15.04.2023

Author Response

Dear IJERPH Editors:  

Thank you for the prompt and comprehensive review of our paper, Age, Loneliness, and Social Media Use in Adults During COVID-19: A Latent Profile Analysis.” We were pleased and greatly appreciate the many complimentary remarks from the reviewers. Additionally, we have carefully considered the Reviewers comments and have revised the original manuscript accordingly. Below, we list each of the Reviewers’ critiques and state how we addressed each point in our revision (in bold). Changes in the manuscript are indicated by red font. 

REVIEWER 2 

Comments and Suggestions for Authors  

I read with interest the manuscript entitled ‘Age, Loneliness, and Social Media Use in Adults During COVID-19: A Latent Profile Analysis’.  

  • I appreciate the authors' idea to approach the relationship between loneliness, age and the use of social media using person-centered analysis. 
  • The manuscript is presented in an organized, systematized manner.  
  • The introduction is clear. However, the authors should structure it:  
  • The paragraph referring to the aim of the study (lines 39-42) should be written after the literature review and related with paragraphs about the study’s approach (line 102) and the hypothesis (line 107, 114);  

RESPONSE: We have edited the Introduction so that the first paragraph ends with a description of the overarching aim of the study and more details regarding aim, approach and hypotheses are stated at the end of the Introduction, following a review of the background literature.  

  • The information about participants (lines 42-47) should be also written after the presentation of previous studies and possible related to the paragraph referring to the variables included in the study (lines 111).  

RESPONSE: We have edited the Introduction to address this point.  

  • Regarding the method, the statistical analysis is elaborated, and the identification of the models five profiles of adults that differed in age, social media use, and loneliness is interesting.  
  • In the results section,  
  • Table 1 – the name of the second column should perhaps be changed, not everything written below represents means and SD (there are also percentages, numbers); 

RESPONSE: We have edited the name of the second column in Table 1 to include % (n).  

  • Table 4 –the specification regarding the letters a, b, c is not clear: according to the legend of the table, it would seem that all values are significant at p< 0.05 and it is not clear why some letters are written in normal font (comparing with the ones written in superscript format)  

RESPONSE: We corrected formatting errors in the original Table 4; now all letters are superscripts. Additionally, the legend was edited to improve interpretation of the superscript letters.  

  • Discussions are clearly presented.  

I congratulate the authors that, through the person-centered approach and identification of the 5 profiles, they emphasized that the relationship between age, loneliness and the use of social media must be viewed from a more nuanced perspective.  

Date: 15.04.2023 

Reviewer 3 Report

Dear Authors,

Thank you for the opportunity to review an interesting article entitled: ‘Age, Loneliness, and Social Media Use in Adults During COVID-19: A Latent Profile Analysis’. The aim of this study was to analyse the possible role of technological barriers in the relationship occurring between SNS use and loneliness during the COVID-19 pandemic. The subject matter is very interesting.

The strengths of the article presented for evaluation are the large sample size, the statistical analyses used, the citation of current literature, as well as the authors' awareness of the strengths and weaknesses of the analyses carried out.

The manuscript contains only a few editorial errors:

[1].  The description of the participants should be in section 2.1, not 3.1.

[2].  The dot at the end of line 159 is redundant.

[3].  Please annotate Cronbach's α for the scales used in their descriptions in lines 159-180.

[4].  If possible, please add ω McDonalds for your study.

[5].  The notation of the designations 'p', 'df', etc., which should be in italics each time, should be standardised.

[6].  There is no legend under Table 4. It is not clear what the letters a, b, c in the subscripts stand for next to the results presented.

[7].  DOI numbers should be completed in the bibliography - items 25, 26, 27, 30.

[8].  The bibliography should be adapted to the requirements of the journal.

Author Response

Dear IJERPH Editors:  

Thank you for the prompt and comprehensive review of our paper, Age, Loneliness, and Social Media Use in Adults During COVID-19: A Latent Profile Analysis.” We were pleased and greatly appreciate the many complimentary remarks from the reviewers. Additionally, we have carefully considered the Reviewers comments and have revised the original manuscript accordingly. Below, we list each of the Reviewers’ critiques and state how we addressed each point in our revision (in bold). Changes in the manuscript are indicated by red font. 

REVIEWER 3  

Comments and Suggestions for Authors 

Dear Authors, 

Thank you for the opportunity to review an interesting article entitled: ‘Age, Loneliness, and Social Media Use in Adults During COVID-19: A Latent Profile Analysis’. The aim of this study was to analyse the possible role of technological barriers in the relationship occurring between SNS use and loneliness during the COVID-19 pandemic. The subject matter is very interesting. 

The strengths of the article presented for evaluation are the large sample size, the statistical analyses used, the citation of current literature, as well as the authors' awareness of the strengths and weaknesses of the analyses carried out. 

The manuscript contains only a few editorial errors: 

[1].  The description of the participants should be in section 2.1, not 3.1. 

RESPONSE: The Participants section (2.1) was expanded to include the description of the participants as well as Table 1. 

[2].  The dot at the end of line 159 is redundant. 

RESPONSE: We have corrected this error.  

[3].  Please annotate Cronbach's α for the scales used in their descriptions in lines 159-180. 

RESPONSE: We report internal reliability data in the Results section under the heading “Psychometrics Properties of the Study Questionnaires,” because we report multiple alphas and other analyses regarding the scales. We reasoned that a reader might appreciate having all of that information in one place. However, if after reading the revised paper, you still disagree, we will move the reliability results in the Methods section.   

[4].  If possible, please add ω McDonalds for your study.  

RESPONSE: We have added McDonalds ω along with Cronbach's α for all reliability analyses. 

[5].  The notation of the designations 'p', 'df', etc., which should be in italics each time, should be standardised.  

RESPONSE: All statistical notations have been checked and standardized.  

[6].  There is no legend under Table 4. It is not clear what the letters a, b, c in the subscripts stand for next to the results presented. 

RESPONSE: We have edited the legend to improve interpretation of the superscript letters.  

[7].   DOI numbers should be completed in the bibliography - items 25, 26, 27, 30.  

RESPONSE: DOI numbers have been added to all items in the bibliography.  

[8].  The bibliography should be adapted to the requirements of the journal. 

RESPONSE: All references have been checked and edited according to the journal requirements.